# The Interaction Relationship of Aroma Components Releasing with Saliva and Chewing Degree during Grilled Eels Consumption

**DOI:** 10.3390/foods12112127

**Published:** 2023-05-25

**Authors:** Xuhui Huang, Huilin Zhao, Renrong Guo, Fei Du, Xiuping Dong, Lei Qin

**Affiliations:** School of Food Science and Technology, National Engineering Research Center of Seafood, Collaborative Innovation Center of Seafood Deep Processing, Dalian Polytechnic University, Dalian 116034, China; xuhuisos@dlpu.edu.cn (X.H.); dxiuping@163.com (X.D.)

**Keywords:** chewing simulation, saliva, aroma components, release, interaction

## Abstract

The interaction perception between aroma and oral chewing during food consumption has always been a hot topic in exploring consumers’ preferences and purchase desires. A chewing simulation system was set to find out the effect of key saliva components and chewing time on odorants released with grilled eel meat. Odor release did not always enhance with the degree of chewing, or the amount of saliva released. The breaking up of the tissue structure of the fish meat by the teeth encourages the release of odorants and the participation of saliva partially blocks this process. The release of pyrazine, alcohol, and acid compounds in grilled eel meat peaked within 20–60 s after chewing. Sufficient exposure of saliva to grilled eel meat will inhibit aromatic, ketone, ester, hydrocarbon, and sulfur compounds release. 3-methyl-2-butanol contributed to the subtle aroma differences that arise before and after eating grilled eel meat. Naphthalene, 2-acetylthiazole, 2-decenal, 2-undecanone, 5-ethyldihydro-2(3H)-furanone were the main odorants released in large quantities in the early stages of eating grilled eel and affected the top note. Consequently, the results provided the odorants information in aroma perception during grilled eel consumption and benefited the objective evaluation of grilled eel product optimization.

## 1. Introduction

Traditional food, bearing the food culture and regional characteristics, has been concerned around the world. Each country has long been committed to the modernization of traditional food processing. As one of the typical traditional fish products, grilled eels attract consumers with their unique flavor. The sensory perception changes of consumers to the flavor quality of grilled eels have always been the focus of its process modernization. Aroma is one of the important indexes for consumers to evaluate the acceptability of grilled eels. Among them, consumers’ sensory perception of food aroma is mainly divided into two stages, one is to smell through the nasal cavity before eating, and the other stage is to feel through the nasal cavity after eating [1]. Due to the complex physiological characteristics of the mouth, it is difficult to achieve a comprehensive analysis of the release of aroma in the process of grilled eel consumption and the composition of odorants in the mouth [2,3]. How to realize the difference in the composition profile of flavor substances in the mouth triggered by different food consumption processes is a research hotspot.

The mouth is like a complex food processing machine, with various components processing food to varying degrees when a person eats, such as the teeth breaking up food, the tongue mixing food, saliva enzymatically diluting food, etc. [2,4,5]. Those processes are essential for digestion and absorption and provide the sensory attributes necessary for consumers. The aroma perception during eating is mainly influenced by the rate and composition of released volatile compounds from food in the mouth [1,5,6]. Chewing usually increases the surface area of food in contact with air in the mouth, which will increase the release of volatile compounds. The destruction of plant and animal tissues during mastication can lead to the production of volatile compounds in the mouth. In addition, saliva can hydrate, dilute, or bond specific volatile compounds through proteins, altering the distribution of volatile compounds between food, saliva, and air, and affecting the diffusion of volatile compounds into the postnasal cavity [6,7]. Mastication factors interact to regulate the release of aroma compounds. It is a complicated and ever-changing process.

Oral processing behaviors, such as chewing time, salivation, and swallowing, have been found to have significant effects on flavor perception during the consumption of foods such as cheese [8], bread [9], and wine [10]. The release rate of flavor compounds during chewing varied between cheese types. The chewing time required to reach the maximum flavor concentration varied as well. The differences in the composition of the food matrix could influence the release of flavor compounds during oral chewing [11]. Saliva content has a significant effect on the release of aromatic compounds in French bread, with saliva diluting and hydrating the aromatic compounds in French bread, thereby slowing their release [12]. A similar relationship between the composition of saliva and the perception of wine aromas was found in studies of wine consumption, where differences in saliva composition led to changes in the formation and release of aromas in wine [13]. However, the complex changes in the oral environment make it difficult to fully explain the effects of different factors on the flavor release of different types of food during oral chewing. In our previous studies, the dynamic detection of characteristic aroma compounds during mastication can be realized by using MS-Nose [14]. However, it is difficult to directly connect the oral cavity with chromatography, leading to the difficulty of comprehensive detection and analysis of aroma compounds in the oral cavity. One of the effective methods to reveal oral processing and flavor perception is to simplify the simulation of the complex oral environment and analyze the changes in the flavor profile in the mouth from the perspective of simple influencing factors [9,15].

Several artificial oral systems have been developed and further improvements to simulate masticatory motion for observing chewing trajectory and its physiological constraints [16]. These models include measuring chewing force and jaw movement related to the physical properties of food [17], simulating bolus formation using a cylindrical chamber driven by a flat mechanical plunger [18], and assessing food texture using a robotic simulator [19]. In these models, focused on reproducing the chewing action and trajectory, it is still difficult to explain the volatile compounds released from different foods during mastication. To simplify the complex environment of the oral cavity, reduce the combined effects of the oral environment and further investigate the effect of a single factor in the oral cavity on the release of volatile flavor substances in food, related studies have investigated the effect of the tongue on the release of food flavor compounds during oral processing using a simplified oral cavity simulator [2,15,20]. Although studies have attempted to unravel the mechanisms of flavor perception during oral mastication of food, the mechanisms are still poorly understood due to individual differences and the complexity of the oral environment. Major issues with the flavor-focused models were dealt with the material design and a sealed chamber having oral and nasal equivalent headspace volume [18,21,22]. The main problems with odor-focused models were inert materials and sealed chamber designs with oral and nasal equivalent headspace volume [18,21,22].

In this paper, the volatile compounds extraction needle, headspace bottle with an inert sealing gasket, and ball milling were combined to construct a simulated oral chewing method that could obtain the volatile composition information in the headspace in real-time. It was aimed to compare and analyze the release regulation of different volatile components under different degrees of food fragmentation and salivary environment via the simulated oral chewing method. The release characteristics and interaction factors of different types of compounds under different chewing conditions were revealed. This study will play an important role in further understanding the molecular mechanism of consumers’ sensory perception during eating.

## 2. Materials and Methods

### 2.1. Grilled Eel

Eels were purchased in the market of Dalian, China. Then the grilled eels were made according to the previous studies [23]. The prepared grilled eel meat samples were vacuum-packed and stored at −80 °C for further testing.

### 2.2. Saliva Collection

Saliva was taken from the volunteers. Sixty volunteers were invited from the laboratory members for oral examination, and 12 volunteers with good oral conditions were finally selected for saliva collection. The selected volunteers ranged in age from 23 to 35 and included both sexes. Saliva collection was referred to a previously reported procedure [24]. Volunteers cleaned their oral cavities with water before saliva collection. The volunteers looked at trays full of grilled eels, smelled the aroma of grilled eels, imagined eating grilled eels, and made a chewing motion with their mouths for 30 s. Then volunteers spat the saliva into a 50 mL centrifuge tube. The collected saliva was put in an ice bath immediately. Each volunteer repeated the process six times, rinsing with water and resting for one minute in between. All collected saliva was mixed and stored at a low temperature for later use.

### 2.3. Oral Mastication Simulation

The oral mastication simulation was performed by a ball mill with an inert treated vial. The inert treated vial had an inert sealing gasket at the lid. There is a pinhole for the SPME extraction needle on the sealing gasket. Three-millimeter zirconium beads were used to break samples. The time of the mastication simulation was accurately timed. The vial was quickly taken out after masticating for 10, 30, 60, 120, and 180 s, and the volatile components in the top space of the vial were quickly absorbed by the extraction needle. Water, artificial saliva, and real saliva were all used to analyze the effects of different saliva components on the release of volatile components during oral mastication simulation.

In the comparison experiment of different chewing stages, the mastication degree was divided via the masticating time. The untreated samples served as positive controls. The samples with pure water, artificial saliva, or real saliva were used as the control for the unchewed samples. The samples chewed for 30 s with pure water, artificial saliva, or real saliva were set to the slightly chewed samples. The samples chewed for 120 s with pure water, artificial saliva, or real saliva were the severely chewed samples. Six replicates were prepared for each comparison sample. The total odorants were absorbed by the extraction needle and identified and quantified via GC-MS.

The preparation of saliva in different concentrations during mastication was diluted with pure water. The dilution ratios were 0, 0.25, 0.5, 0.75, and 1. Saliva of different concentrations was mixed with the samples and chewed for 60 s, respectively. Six replicates were prepared for each comparison sample. After chewing, the odorants were absorbed by the extraction needle and identified and quantified via GC-MS.

In the aroma profile analytical experiment, samples for different chewing times and saliva concentrations were prepared as above. Pure water containing no inorganic ions or enzymes, artificial saliva containing only inorganic ions, and real saliva containing inorganic ions and enzymes were used in the experiments. Six replicates were prepared for each comparison sample. Odorants were also absorbed by the extraction needle and identified and quantified via GC-MS(O).

### 2.4. Volatile Compounds Identification

Volatile compounds detection and identification were referred to the previous study [25]. Fiber containing volatile compounds was immediately desorbed in the GC injection port at 250 °C for 2 min. The initial oven temperature was 35 °C and was raised to 280 °C at 5 °C/min after holding for 3 min. The mass scanning range of gas chromatography-mass spectrometry was 45–450. The key odorant identification was used NIST14 and Wiley11 library, the calculated RI of the volatile compounds, and odor sniffing via GC–MS(O) that are referred to the previous study [26]. 

### 2.5. Volatile Compounds Quantitation

The multi-isotope internal standards combined with the external standards method established in previous studies were used for the key aroma compounds quantitation [14]. The stable isotope internal standards, n-nonan-d20, n-dodecane-d26, n-nonadecane-d40, n-tridecane-d28, and n-hexadecane-d34 were used as quantitative internal standards. Due to the difference in the length of the carbon chain, the retention time of the stable isotope internals was regularly distributed, covering the retention time range of all the aroma compounds to be quantified. The selection of a quantitative internal standard was mainly based on the similarity of retention time between the internal standard and the compounds. Standard curves with eight concentration points were prepared. Concentrations ranged from 1 to 500 ng/mL. The content of aroma compounds was calculated by standard curve.

### 2.6. Statistical Analysis

Microsoft Office 2016 (Microsoft, Beijing, China), XLSTAT 2019 (Lumivero, Denver, CO, USA), and SPSS v 24.0 (IBM, Armonk, New York, USA) were used for analyzing data, drawing figures, and combining figures. Significance analysis was conducted by SPSS v 24.0. Box-plot analysis was performed using XLSTAT 2019. The heat map of multidimensional interactive analysis was performed using the OmicStudio tools at https://www.omicstudio.cn/tool and accessed on 1 October 2022.

## 3. Results and Discussion

### 3.1. Interaction between Saliva and Volatiles Release during Mastication

The effects of different chewing stages and saliva components on the release of aroma components of grilled eel were observed through the chewing simulation of the fish-eating process. As shown in Figure 1, by measuring the total amount of volatile components released at the same time, the volatile components released from grilled eel under different chewing degrees and under different saliva simulation systems was compared and analyzed. The total amount of odors released by the control samples was higher than that of the unchewed samples. It was also higher than most chewable samples. The control samples had a total odor intensity of over 75 μg/mL, while the unchewed samples had a total odor intensity of around 50 μg/mL. The control group was directly simulated oral mastication without any type of saliva added. It indicated that the teeth can effectively promote the release of volatile compounds by breaking the tissue structure of fish meat [27,28].

By comparing the differences in the release of volatiles in grilled eel meat under different saliva systems, it was found that deep chewing could increase the release of volatiles in fish. The release regularity was affected by the components of saliva. When simulated with pure water instead of saliva, it was found that the release of volatiles decreased slightly with chewing samples compared to non-chewing samples. When there was no enzyme, ions, or other substances in saliva, it played a role in diluting and hindering the release of volatiles in fish meat. When pure water was substituted by artificial saliva which was constituted via main ionic solutions in saliva without enzymes, different types and concentrations of ions, such as Ca^2+^, K^+^, Na^+^, etc., will affect the three-phase balance of volatile components in fish tissue, saliva, and headspace air in the mouth [6,7]. As can be seen from Figure 1B, in the presence of artificial saliva, volatiles in fish were released in large quantities nearly 90 μg/mL after deep chewing. The amount released was greater than the amount of tissue broken through the teeth alone. It indicated that the ion concentration of saliva was important for the release of aroma compounds in grilled eel meat during chewing.

Comparing the difference between real saliva and artificial saliva on the volatile components of fish, it was found that in the initial simulated chewing stage, the samples treated with real saliva released more volatiles than the samples with artificial saliva. At the deep chewing stage, the release of volatiles in the samples treated with real saliva increased at a much lower rate and amount than in the samples treated with artificial saliva. According to this phenomenon, it can be inferred that in the initial stage of chewing, the enzymes in saliva mainly act on the tissue structure of fish meat. After deep chewing and stirring, the enzymes in saliva would also combine with the volatiles in fish meat to form the bolus and slowed down the release of volatiles [6].

### 3.2. Dynamic Changes of Volatile Components during Mastication under Different Salivary Systems

The saliva in the mouth is very complex. To study the interaction between saliva and volatile components in grilled eel meat during mastication, the dynamic release of different volatile components was compared from a simple system to a complex system. As shown in Figure 2, nine types of volatile compounds were found to release in the simulated process of chewing grilled eel meat, including pyrazine compounds, sulfur compounds, alcohol compounds, aromatic compounds, hydrocarbons, aldehydes, ketones, esters, and acids.

For pyrazine compounds, the volatiles release lows were different in the simulated fish meat chewing process under the three saliva systems. The release of pyrazine compounds in the simulated process of chewing grilled eel with the three saliva systems all had chewing time points for the highest release. The release of pyrazine compounds in the artificial saliva system was like that in real saliva. The release of pyrazine compounds initially increased and then fell back. The pyrazine compounds in grilled eel under the artificial saliva system reached the maximum release ratio of 1.9 after chewing for 60 s, while the real saliva system reached the maximum release ratio of 1.6 after chewing for 30 s. In the pure water system, the release of pyrazine compounds in grilled eel meat decreased with chewing. However, the release of pyrazine compounds in the whole process was greater than that before chewing.

The overall release of alcohol compounds in grilled eel after chewing was also higher than that of grilled eel without chewing. In the first 30 s of the mastication simulation, the release of alcohol compounds showed a downward trend regardless of the saliva systems. It might be caused by the dilution of water [29]. With the further continuation of chewing, in the pure water system, the alcohol compounds release in grilled eel showed a steady trend, while in the artificial saliva and real saliva systems were increased. Both the alcohol compounds release in the artificial saliva and real saliva systems reached a peak at 60 s of chewing with the release ratio of 1.3 and 2.1, respectively. However, the changes between the two were different in the later stage. The release amount of grilled eel treated with artificial saliva was almost unchanged when reach the peak value while that treated with real saliva was decreased. Under the complex protein and ion of real saliva, the release of alcohol compounds would also be restricted with the continuous breaking and mixing of chewing.

The total release of sulfur-containing compounds in grilled eel during chewing was lower than that of grilled eel without chewing. When real saliva was added to simulate the chewing process, the release of sulfur-containing compounds was almost unchanged, with a slight decrease but no significant change. The release law of sulfur-containing compounds was the same in the simulated mastication process of adding water and artificial saliva. Their release peak appeared at 60 s of mastication. The release of sulfur-containing compounds was higher in the pure water system. This may be because the water and artificial saliva systems were simpler than the real saliva system, especially since the water system can simply promote or inhibit the release of compounds by being hydrophilic [6,7]. However, for the complex real salivary system, there was a combination of promotion and inhibition. The combined effect was balanced and weakly correlated with the duration of chewing.

For acid compounds, in the simulated chewing process with real saliva or water added, the release pattern of acid compounds was similar. The number of acid compounds was increased compared with that of unchewed grilled eel. In the mastication process with the addition of artificial saliva, although the release of acid compounds showed similar changes, the changes were weaker and somewhat delayed. Meanwhile, the release amount in the first 60 s of mastication was lower than that without mastication. The artificial saliva system has more ions than the water system and lacks many proteases compared to the real saliva system. It can be indicated that the release of acid compounds during mastication may be closely related to ions in saliva.

The overall releases of aromatic compounds, aldehydes, hydrocarbons, ketones, and esters were similar in the simulated chewing process. The release of aromatic compounds, aldehydes, hydrocarbons, ketones, and esters during simulated oral chewing of grilled eel with real saliva and water was lower than that without chewing and decreased with chewing. For the simulated oral chewing process with artificial saliva, the release peaks of these compounds appeared at 60 s. Except for aromatic compounds, their release peaks were higher than that of grilled eel meat without chewing.

Based on the results above, the saliva concentration was further changed to observe the effect on the release of volatile compounds. As shown in Figure 3, the effects of real saliva and artificial saliva concentration changes on the release of various volatile compounds were similar during the simulated oral chewing of grilled eel meat. The release rules of various volatile compounds in simulated oral chewing were also similar. There was an optimal saliva concentration for release. For aromatic compounds, sulfur-containing compounds, hydrocarbons, and ketones, the relative release ratios were all less than 1. It suggested that the release of these compounds in grilled eel meat would be reduced during mastication due to saliva. Changing the concentration of saliva could reduce its release inhibition effect on volatile release. The effect depends on the amount of water added. For aromatic compounds, hydrocarbons, and ketones, their release peaks were all at 50% salivary concentration, while the release peak of sulfur-containing compounds was at 25% salivary concentration. Saliva concentration depends on the proportion of pure water mixed with pure saliva. Consumption of grilled eel meat with a small amount of water may be able to effectively maintain the release of these compounds and help the aroma perception. This also supported the view that different levels of saliva dilution can cause changes in aroma perception [6]. For pyrazines, alcohols, esters, acids, and aldehydes, oral chewing can promote the release of these compounds in grilled eel meat. Changing the concentration of saliva can also enhance the release of these compounds [6,30]. 

### 3.3. The Characteristic Aroma Profile of Grilled Eel Meat during Chewing Simulation

Through the detection and perception of volatile compounds released from grilled eel meat during simulating the chewing process, 34 characteristic aroma compounds were selected. As shown in Figure 4, the characteristic aroma contour was drawn according to normalized OAV. With the increase in chewing time, the contribution of some characteristic aroma compounds increased. However, the release trend of different compounds did not increase or decrease gradually with the extension of chewing time. Some characteristic aroma compounds had certain release peaks. Although there was a difference in the release rate of characteristic aroma compounds before and after chewing, the difference was not obvious from the overall contour. It indicated that the orthonasal aroma perception was similar to the retronasal aroma perception when eating grilled eel [1,3].

During simulated oral chewing of grilled eel meat, 3-methyl-2-butanol was the main characteristic aroma compound that changed significantly. 3-methyl-2-butanol presented a fresh aroma like fruit. Before simulated chewing, the odor contribution value of 3-methyl-2-butanol to the overall aroma of grilled eel meat was greater. However, it was not detected after simulated chewing and had no contribution to the characteristic aroma of grilled eel meat. It suggested that 3-methyl-2-butanol may play a larger role in the subtle flavor differences that arise when grilled eel was eaten. In addition, the release of aldehyde, ketone, and alcohol compounds, such as nonanal (such as citrus odor), 2-decenal (such as rose odor), decanal (such as orange peel odor), 2-nonenal (such as cucumber odor), dihydro-5-propyl-2(3H)-furanone (such as coconut odor), 2-undecanone (such as fruity odor), 1-octen-3-ol (such as mushroom odor), etc., was also decreased after chewing. On the contrary, acid compounds, such as 3-methyl-butanoic acid, octadecanoic acid, and tetradecanoic acid were increased after chewing. The increase in acid compounds changed the aroma perception of grilled eel during eating. These acids also had a sour taste. The sensory interaction will further amplify the characteristic flavor of grilled eel [31,32].

Differences in the proportion of saliva components can lead to different effects on the release of different compounds during eating grilled eel meat. Changing saliva concentration can appropriately alter the enhancement or inhibition of the release of certain characteristic compounds. As shown in Figure 5, the concentration of saliva had a lesser effect on the release of characteristic aroma compounds such as nonanal, 2-decenal, octadecanoic acid, etc., than chewing duration. When the saliva concentration was at a low value, the release of 1-methyl-naphthalene, 2-methyl-naphthalene, phenylethyl alcohol, benzothiazole, and phenanthrene was lower than that under high saliva concentration. All these compounds contained a benzene ring structure. It can be indicated that there was a certain correlation between the saliva concentration during chewing and the structure of characteristic compounds released during eating grilled eel meat [6,29,30]. Saliva was complex, rich in a large number of ions, different enzymes, and protein components, such as amylase, proline-rich proteins, mucin proteins, etc. [6]. A change in concentration led to a decrease in its ion buffering, which in turn led to a change in its pH. This change affected the distribution of these compounds in fish meat, saliva, and oral air. In addition, it was possible that enzymes in saliva also changed the structure of the fish meat fiber to promote the release of these compounds. As the concentration was diluted, the effect gradually decreased, resulting in this phenomenon.

### 3.4. Interaction between the Release of Characteristic Aroma Compounds and Key Factors of Simulated Mastication

Based on the above results, the duration of chewing and the amount of saliva can significantly affect the release of characteristic aroma compounds during the consumption of grilled eel meat. The enzymes and ions in saliva were one of the main influencing factors. To further explain the relationship between the 34 key aroma compounds screened and chewing time, enzymes, and ions in saliva, the results were processed through multidimensional interactive analysis, as shown in Figure 6. The release of key aroma compounds, such as naphthalene, 2-acetylthiazole, 2-decenal, 2-undecanone, 5-ethyldihydro-2(3H)-furanone, etc., was negatively correlated with mastication. It indicated that these compounds were heavily released in the early stages of grilled eel consumption. They were contributed to the top note during chewing grilled eel. On the contrary, the release of key aroma compounds, hexadecanoic acid, methyl ester, phenylethyl alcohol, 3-methyl-butanoic acid, and octadecanoic acid was positively correlated with mastication. They contributed to the aroma of grilled eel meat after a long time of chewing. In addition, the release changes of decanal, 2-nonenal, 2,6-diethyl-pyrazine, 2-undecenal, nonanal, etc. had little correlation with the degree of mastication, which may be the main compounds’ affected basic note in the process of eating grilled eel. 

The properties of saliva were also correlated with the release of characteristic aroma compounds from the grilled eel meat. As shown in Figure 6, the release of phenanthrene, 2-methoxy-phenol, benzothiazole, 3-methyl-butanoic acid, and octadecanoic acid during eating grilled eel meat was related to saliva. The release of 3-methyl-butanoic acid was directly related to the concentration of enzyme in saliva while the release of phenanthrene, 2-methoxy-phenol, benzothiazole, etc., was due to the ions in saliva. The release of saliva during chewing and the effect of the released saliva on the structure of grilled eel meat after crushing may directly change the three-phase distribution of these compounds in the fish, saliva, and the oral air, which promoted the release of these compounds [6,22]. The release of benzaldehyde, 2-ethyl-1-hexanol, and naphthalene was negatively correlated with the release of saliva. It was related to the fact that large molecular compounds such as proteins in saliva form bolus with these characteristic compounds as chewing progresses and inhibited their release.

## 4. Conclusions

The oral mastication simulation combined with the volatile compound extraction needle, headspace bottle with inert sealing gasket, and ball milling can be employed to obtain the key odorants information and reflect the process of eating grilled eel meat. The release of food in the mouth was reduced by enzymes in the saliva. The enzymes in saliva combined with the volatiles in fish meat to form the supramolecular complex and slowed down the release of volatiles. Ions in saliva, Ca^2+^, K^+^, Na^+^, etc., also affected the three-phase balance of volatile components in fish tissue, saliva, and headspace air in the mouth to increase volatile release. Pyrazine, alcohol, and acid compounds in grilled eel meat were the first to be released during chewing and form the unique aroma. The sufficient mix of saliva and fish inhibited the release of aromatic, ketone, ester, hydrocarbon, and sulfur compounds release. Drinking a small amount of pure water to dilute the saliva during eating grilled eel meat helped to release aromatic, hydrocarbons, ketones, and sulfur compounds and attain a different aroma perception. 3-methyl-2-butanol was one of the key compounds that result in the different aroma sensations before and after grilled eel meat consumption. Naphthalene, 2-acetylthiazole, 2-decenal, 2-undecanone, 5-ethyldihydro-2(3H)-furanone mainly affected the top note in the process of eating grilled eel while decanal, 2-nonenal, 2,6-diethyl-pyrazine, 2-undecenal, nonanal mainly affected the basic note. Consequently, this especially elucidated the main odorants in different grilled eel consumption stages and gave a guide to the process of fish products’ improvement.

## Figures and Tables

**Figure 1 foods-12-02127-f001:**
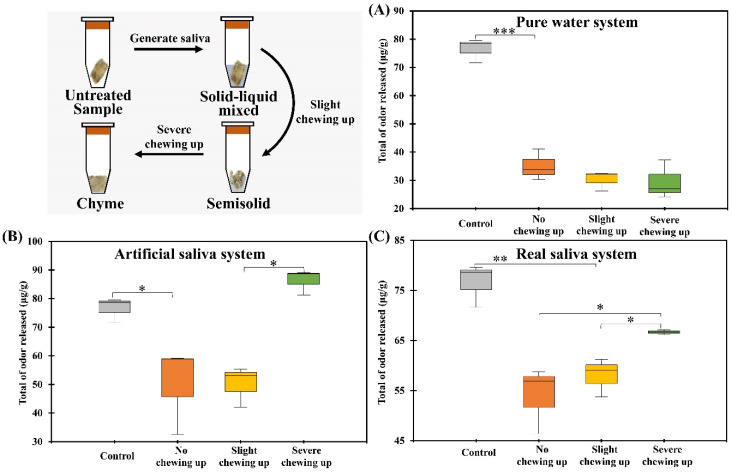
Change of aroma release under different chewing systems. (**A**) was the Box-plot of total odor release during grilled eel meat chewing under pure water system. (**B**) was the Box-plot of total odor release during grilled eel meat chewing under artificial saliva system. (**C**) was the Box-plot of total odor release during grilled eel meat chewing under real saliva system. The significance level: * indicated *p* < 0.05; ** indicated *p* < 0.01; *** indicated *p* < 0.001.

**Figure 2 foods-12-02127-f002:**
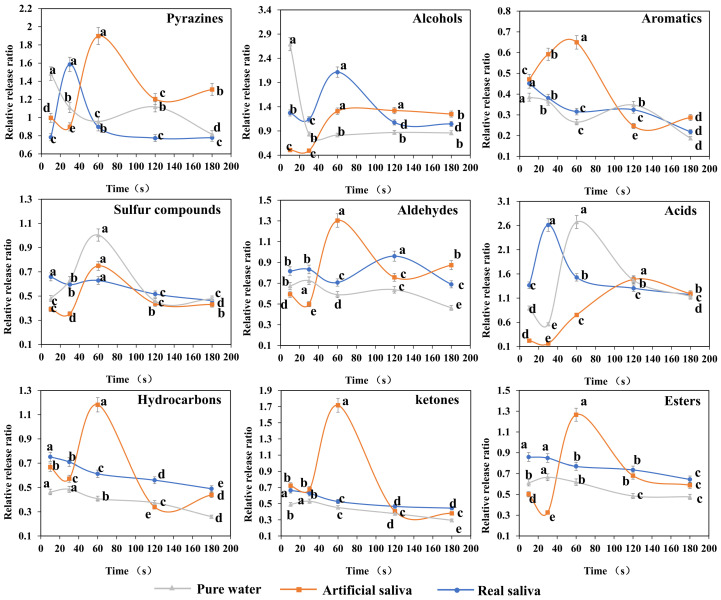
Time dynamic curves of the release of different volatile components during mastication of grilled eel meat under different salivary systems. Relative release ratios were obtained by comparing the corresponding amounts of various compounds before and after chewing. Values > 1 indicated that the release of volatiles after chewing was increased than that before chewing. Different letters in the same curve indicate significant differences (*p* < 0.05).

**Figure 3 foods-12-02127-f003:**
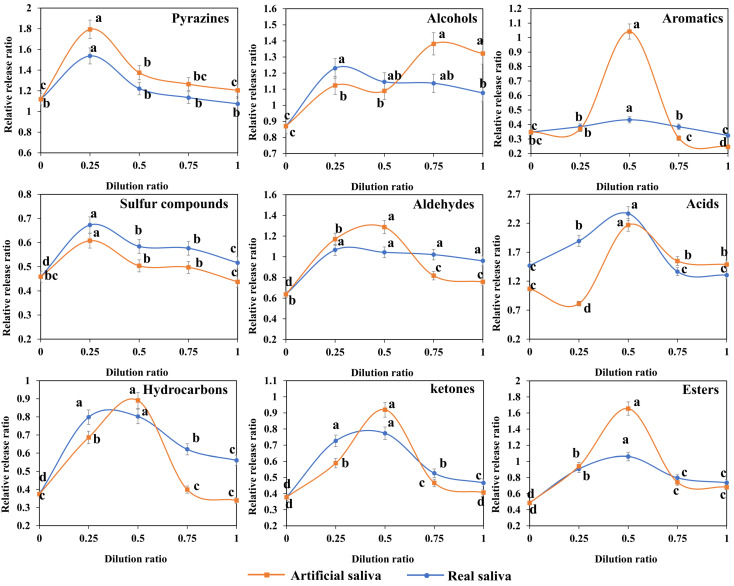
Release curves of different volatile components during mastication of grilled eel meat under different salivary concentrations. The dilution ratio of saliva indicated the proportion of saliva in the system: 0 means no saliva, while 1 means undiluted saliva. Relative release ratios were obtained by comparing the corresponding amounts of various compounds before and after chewing. Values > 1 indicated that the release of volatiles after chewing was increased than that before chewing. Different letters in the same curve indicate significant differences (*p* < 0.05).

**Figure 4 foods-12-02127-f004:**
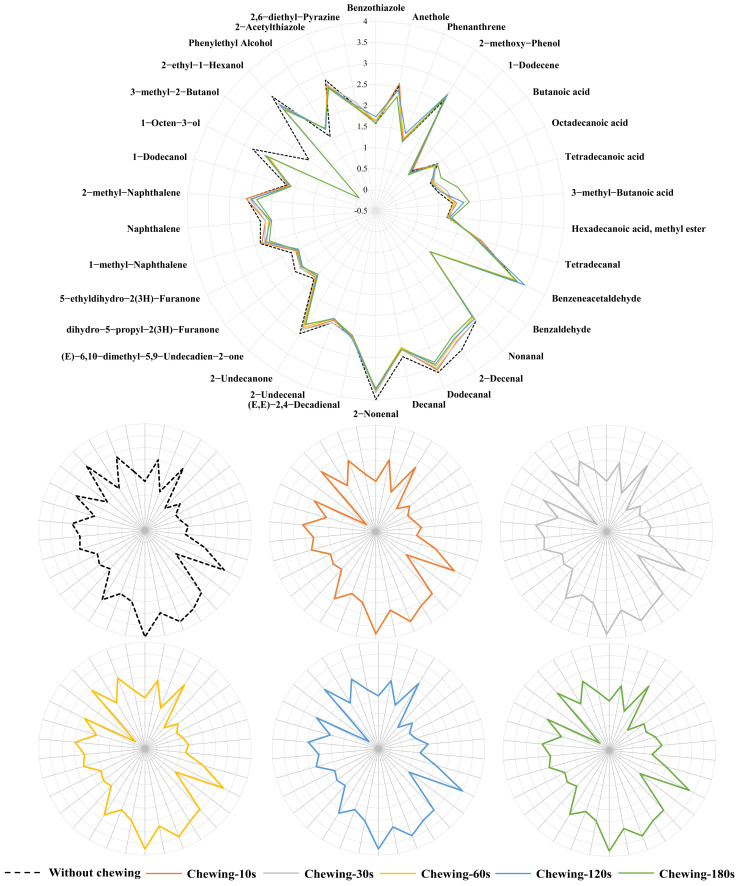
The characteristic aroma profile of grilled eel meat during different chewing simulation times with real saliva. The aroma profile was drawn by the normalized data of the OAV value raised to the 0.1 power.

**Figure 5 foods-12-02127-f005:**
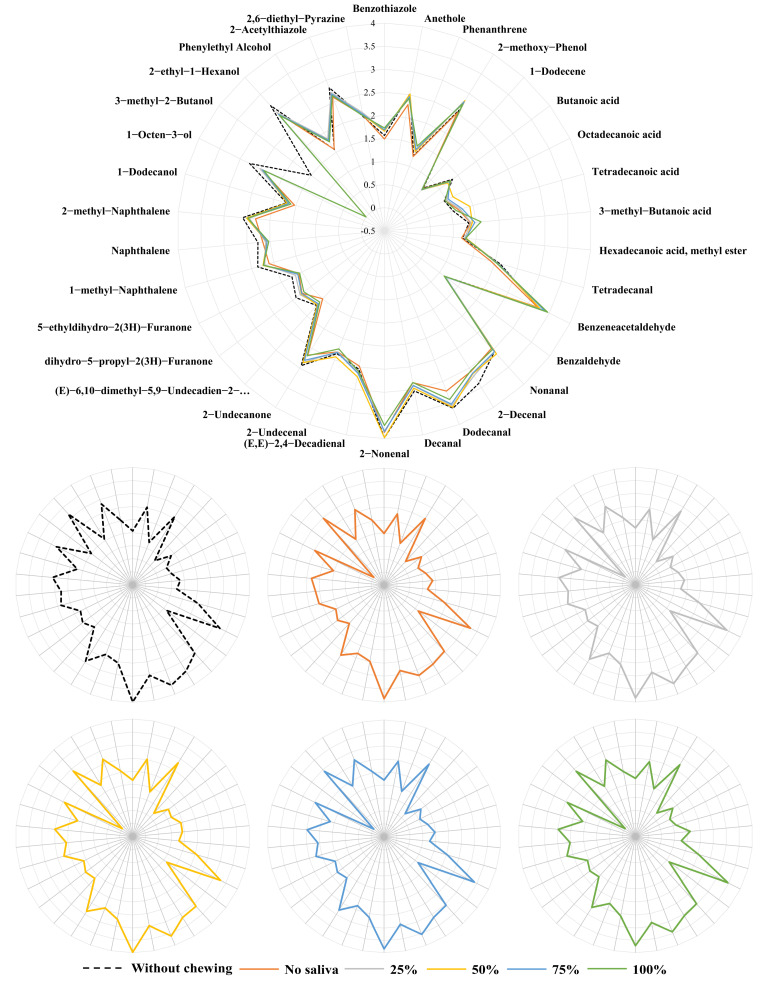
The characteristic aroma profile of grilled eel meat under different real saliva concentrations during chewing simulation. The aroma profile was drawn by the normalized data of the OAV value raised to the 0.1 power.

**Figure 6 foods-12-02127-f006:**
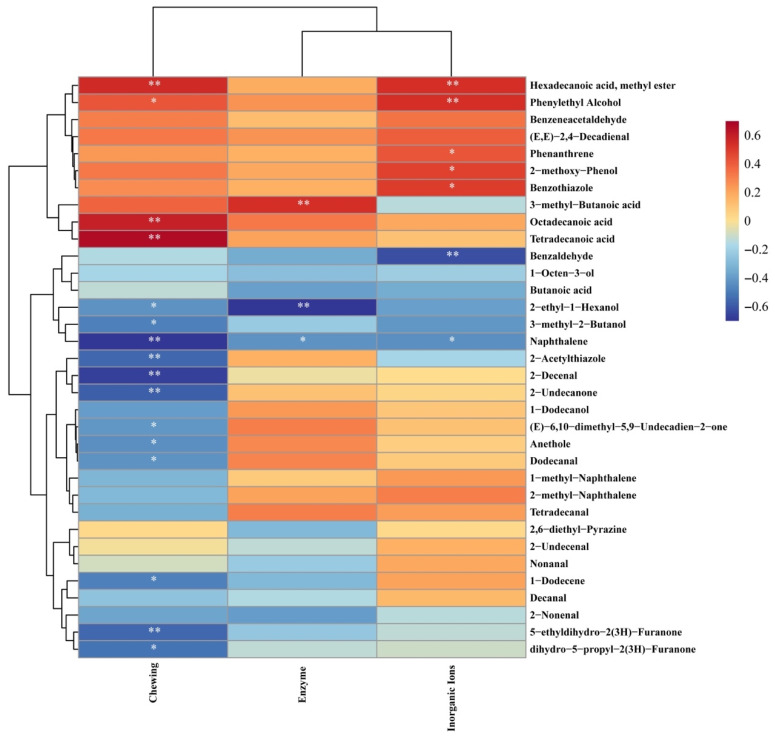
The heat map of multidimensional interactive analysis between characteristic aroma compounds and key factors of simulated mastication. Each colored cell on the heat map corresponds to a correlation value between aroma compounds and chewing, enzyme, or inorganic ions. The significance level: * indicated *p* < 0.05; ** indicated *p* < 0.01.

## Data Availability

Data is contained within the article.

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
