# Peer review of "The Interaction Relationship of Aroma Components Releasing with Saliva and Chewing Degree during Grilled Eels Consumption"

_foods, 2023, doi:10.3390/foods12112127_

Round 1
Reviewer 1 Report
The manuscript submitted by Huang et al. studied the effect of saliva components and chewing time on odorants released with grilled eel meat. This research shows important information related to the chewing process of grilled eel meat. Unfortunately, making specific observations and comments on the manuscript is difficult if it lacks numbering. For this reason, comments, suggestions, and observations are only made in a general way.
In my opinion, for this manuscript to be considered, authors must make significant changes:
Title
The title does not appropriately reflect the research conducted. Please, revise the title of the manuscript to reflect better the focus of the research, including the food under study. An appropriate title would make it easier for readers to understand the purpose of the study before diving into the details.
Abstract
The authors state: “the odorants releasing was not always positively correlated with the degree of chewing and the amount of saliva released” However, according to the results, no correlation study was carried out between the degree of chewing and the amount of saliva released. Therefore, this statement needs to be modified.
It suggested a general conclusion at the end of the abstract focused on the importance of the information obtained to understand the chewing process of grilled eel meat and how this information can be used to optimize the product and increase its consumption.
Introduction
The study was carried out on grilled eel meat, and the authors should highlight the importance and why this food was selected.
Materials and Methods
The authors designed a set of experiments including:
- The effect of different chewing stages and saliva components on the release of aroma components of grilled eel meat.
- The effect of salivary systems on the release of different volatile components during mastication of grilled eel meat.
- The effect of salivary concentration on different volatile components during mastication of grilled eel meat.
- The effect of chewing time on the aroma profile of grilled eel meat.
- The effect of saliva concentration on aroma profile during chewing grilled eel meat.
- The relation between characteristic aroma compounds and key factors of simulated mastication (chewing, enzymes, and inorganic ions).
All these experiments were interesting and important information was obtained. However, the materials and methods section needed to describe these experiments correctly. Therefore, a brief description of each experiment is necessary to ensure repeatability. It is also necessary to include the number of samples analyzed, the statistical analysis performed, the means comparisons, if so, the p-value, etc.
Results and discussion
3.1 section: The authors state: “The total amount of odors released by the control samples was higher than…”, “It was also higher than…”, “it was found that the release of volatiles decreased…”, “The amount released was greater than…”, etc. These types of statements must be based on statistical data analysis and accompanied by the p-value obtained. The materials and methods section does not specify how the data analysis was carried out and how the means comparison was performed. The same observation is in the subsequent sections of the results.
Fig 1. The figure is blurred, and it is not easy to read. The quality of this figure needs to be improved. Also, the y-axis scale should be uniform; for example, from 30 to 100. Authors should include the differences between means and their significance, as well as the number of samples analyzed.
Fig 2 and Fig 3. Please specify the number of samples analyzed, the meaning of the bars in the averages, and the p-values resulting from the statistical analysis.
Fig 6. Please include the meaning of the colors and symbols *, **
Conclusions
Much of the information in the conclusions is more of a summary of the results; please rewrite the conclusions.

Author Response
Response to Reviewer #1
1) The title does not appropriately reflect the research conducted. Please, revise the title of the manuscript to reflect better the focus of the research, including the food under study. An appropriate title would make it easier for readers to understand the purpose of the study before diving into the details.
Response: Thanks for your good suggestion. The title has been revised.
“The Interaction Relationship of Aroma Components Releasing with Saliva and Chewing Degree during Grilled Eels Consumption”
2) The authors state: “the odorants releasing was not always positively correlated with the degree of chewing and the amount of saliva released” However, according to the results, no correlation study was carried out between the degree of chewing and the amount of saliva released. Therefore, this statement needs to be modified.
Response: Thanks for your good suggestion. The sentence has been revised.
“Odor release did not always enhance with the degree of chewing or the amount of saliva released.”
3) It suggested a general conclusion at the end of the abstract focused on the importance of the information obtained to understand the chewing process of grilled eel meat and how this information can be used to optimize the product and increase its consumption.
Response: Thanks for your good suggestion. A general conclusion at the end of the abstract has been revised.
“The results provided the odorants information in aroma perception during grilled eel consumption and benefited to the objective evaluation of the grilled eel products optimization.”
4) The study was carried out on grilled eel meat, and the authors should highlight the importance and why this food was selected.
Response: Thanks for your good suggestion. The information on grilled eels has been added. The importance of why selecting grilled eels has been explained.
“Traditional food, bearing the food culture and regional characteristics, has been a concern around the world. Each country has long been committed to the modernization of traditional food processing. As one of the typical traditional fish products, grilled eels attract consumers with their unique flavor. The sensory perception changes of consumers to the flavor quality of grilled eels have always been the focus of its process modernization.”
5) The authors designed a set of experiments including:
1.The effect of different chewing stages and saliva components on the release of aroma components of grilled eel meat.
2.The effect of salivary systems on the release of different volatile components during mastication of grilled eel meat.
3.The effect of salivary concentration on different volatile components during mastication of grilled eel meat.
4.The effect of chewing time on the aroma profile of grilled eel meat.
5.The effect of saliva concentration on aroma profile during chewing grilled eel meat.
6.The relation between characteristic aroma compounds and key factors of simulated mastication (chewing, enzymes, and inorganic ions).
All these experiments were interesting and important information was obtained. However, the materials and methods section needed to describe these experiments correctly. Therefore, a brief description of each experiment is necessary to ensure repeatability. It is also necessary to include the number of samples analyzed, the statistical analysis performed, the means comparisons, if so, the p-value, etc.
Response: Thanks for your good suggestion. Detailed information on these experiments has been added.
“In the comparison experiment of different chewing stages, the mastication degree was divided via the masticating time. The untreated samples served as positive controls. The samples with pure water, artificial saliva, or real saliva were used as the control for the unchewed samples. The samples chewed for 30 s with pure water, artificial saliva, or real saliva were set to the slightly chewed sample. The samples chewed for 120 s with pure water, artificial saliva or real saliva were the severely chewed samples. Six replicates were prepared for each comparison sample. The total odorants were absorbed by the extraction needle and identified and quantified via GC-MS.
The preparation of saliva in different concentrations during mastication was diluted with pure water. The dilution ratios were 0,0.25,0.5,0.75 and 1. The saliva of different concentrations was mixed with the samples and chewed for 60 seconds, respectively. Six replicates were prepared for each comparison sample. After chewing, the odorants were absorbed by the extraction needle and identified and quantified via GC-MS.
In the aroma profile analytical experiment, samples for different chewing times and saliva concentrations were prepared as above. Pure water containing no inorganic ions or enzymes, artificial saliva containing only inorganic ions, and real saliva containing inorganic ions and enzymes were used in the experiments. Six replicates were prepared for each comparison sample. Odorants were also absorbed by the extraction needle and identified and quantified via GC-MS(O).”
6) 3.1 section: The authors state: “The total amount of odors released by the control samples was higher than…”, “It was also higher than…”, “it was found that the release of volatiles decreased…”, “The amount released was greater than…”, etc. These types of statements must be based on statistical data analysis and accompanied by the p-value obtained. The materials and methods section does not specify how the data analysis was carried out and how the means comparison was performed. The same observation is in the subsequent sections of the results.
Response: As shown in Fig. 1, the discussion on the total amount of odors was based on the quantitative results of the odorant content of different samples. Details have been added in the Materials and Methods. Microsoft Office 2016 (Microsoft, China), XLSTAT 2019 (Addinsoft, France), and SPSS v 24.0 (IBM, USA) were used for analyzing data, drawing figures, and combining figures. The statistical differences between different data were determined by SPSS v 24.0. Boxplot was made via XLSTAT 2019. Besides, the difference analysis results have been added in Fig. 1.
7) Fig 1. The figure is blurred, and it is not easy to read. The quality of this figure needs to be improved. Also, the y-axis scale should be uniform; for example, from 30 to 100. Authors should include the differences between means and their significance, as well as the number of samples analyzed.
Response: The resolution size of the graph in the paper may be compressed due to the submission system. The resolution of Fig. 1 has been improved and significance analysis has been added. Due to the wide variation in the total odor content of different samples, to see the differences more intuitively between the same comparison group, the Y-axis was set according to the size of the unified comparison group. Therefore, the y-axis scale did not be uniform between different comparison groups.
8) Fig 2 and Fig 3. Please specify the number of samples analyzed, the meaning of the bars in the averages, and the p-values resulting from the statistical analysis.
Response: Thanks for your good suggestion. Six replicates were made for each sample. Details have been added in the Materials and Methods. The bars in the figure are error lines. It was conducted by analyzing the difference of the measured values between different replicates. The p-values have been added as well.
9) Fig 6. Please include the meaning of the colors and symbols *, **
Response: Thanks for your good suggestion. Each colored cell on the heat map corresponds to correlation value between aroma compounds and chewing, enzyme, or inorganic ions. The significance level: * indicated p<0.05; ** indicated p<0.01. Details have been added in the Figure 6 legend.
10) Much of the information in the conclusions is more of a summary of the results; please rewrite the conclusions.
Response: The conclusion has been revised.
“The oral mastication simulation combined with volatile compounds extraction needle, headspace bottle with inert sealing gasket, and ball milling can be employed to obtain the key odorants information and reflect the process of eating grilled eel meat. The release of food in the mouth was reduced by enzymes in the saliva. The enzymes in saliva combined with the volatiles in fish meat to form the supramolecular complex and slowed down the release of volatiles. Ions in saliva, Ca2+, K+, Na+, etc., also affected the three-phase balance of volatile components in fish tissue, saliva, and headspace air in the mouth to increase volatile release. Pyrazine, alcohol, and acid compounds in grilled eel meat were the first to be released during chewing and form a unique aroma. The sufficient mix of saliva and fish inhibited the release of aromatic, ketone, ester, hydrocarbon, and sulfur compounds release. Drinking a small amount of pure water to dilute the saliva during eating grilled eel meat helped to release aromatic, hydrocarbons, ketones, and sulfur compounds and get a different aroma perception. 3-methyl-2-butanol was one of the key compounds that result in the different aroma sensations before and after grilled eel meat consumption. Naphthalene, 2-acetylthiazole, 2-decenal, 2-undecanone, 5-ethyldihydro-2(3H)-furanone mainly affected the top note in the process of eating grilled eel while decanal, 2-nonenal, 2,6-diethyl-pyrazine, 2-undecenal, nonanal mainly affected the basic note. Consequently, this especially elucidated the main odorants in different grilled eel consumption stages and gave a guide to the process of fish products improvement.”

Reviewer 2 Report
<Introduction>
Since it is a general, clearly indicate the purpose with the experiment
<Method>
How many repetitions of 2.3 & 2.4 ? Describe the number of repetitions that can explain the statistical significance.
What kind of fibers are used in SPME ? References are not readily available for me, so it is a good to include them
Regarding the quantitative analysis method in 2.5, describe the method in more detail.
In 2.6, clearly describe what kind of analysis you performed.
<Results>
Is the “the “number” of volatile components released'' in line 5 of 3.1 a mistype of “amount''? Figure 1 is data based on quantity, so if it is a number, it is better to indicate such as data not shown.
Figure 1, no statistical treatment was done, and it is unclear whether the difference is significant. Describe the number of times, statistical processing method, significance level in legend.
3.1 Line 5 from the end, "the enzymes in saliva mainly act on the tissue structure of fish meat [7]", is the citation appropriate?
3.1, page 5, line 2 from the end and 3.3, page 13, line 3 from the end, "the supramolecular complex" simply seems to be more appropriate as " bolus ".
3.2, page 8, line 6 from the bottom, ”Consumption of grilled eel meat with a small amount of water may be able to effectively maintain the release of these compounds and help the aroma perception . ” How about quoting or describing the basis, and it will become clear whether 25% is a realistic set value
Figure 4. Conditions other than chewing time are unknown, so please describe them in legend
3.3, page 10, last line, ”The increase in acid compounds not only changed the aroma perception of grilled eel during eating but also its taste.” You mention the taste, but do you have sensory evaluation data? If not, please indicate data not shown.
3.3, page 11, line 6, “Adjusting saliva concentration'', it is difficult to adjust the concentration of saliva in humans, so I feel that an expression such as Changing is more suitable (only English expression)

<Introduction>
Since it is a general, clearly indicate the purpose with the experiment
<Method>
How many repetitions of 2.3 & 2.4 ? Describe the number of repetitions that can explain the statistical significance.
What kind of fibers are used in SPME ? References are not readily available for me, so it is a good to include them
Regarding the quantitative analysis method in 2.5, describe the method in more detail.
In 2.6, clearly describe what kind of analysis you performed.
<Results>
Is the “the “number” of volatile components released'' in line 5 of 3.1 a mistype of “amount''? Figure 1 is data based on quantity, so if it is a number, it is better to indicate such as data not shown.
Figure 1, no statistical treatment was done, and it is unclear whether the difference is significant. Describe the number of times, statistical processing method, significance level in legend.
3.1 Line 5 from the end, "the enzymes in saliva mainly act on the tissue structure of fish meat [7]", is the citation appropriate?
3.1, page 5, line 2 from the end and 3.3, page 13, line 3 from the end, "the supramolecular complex" simply seems to be more appropriate as " bolus ".
3.2, page 8, line 6 from the bottom, ”Consumption of grilled eel meat with a small amount of water may be able to effectively maintain the release of these compounds and help the aroma perception . ” How about quoting or describing the basis, and it will become clear whether 25% is a realistic set value
Figure 4. Conditions other than chewing time are unknown, so please describe them in legend
3.3, page 10, last line, ”The increase in acid compounds not only changed the aroma perception of grilled eel during eating but also its taste.” You mention the taste, but do you have sensory evaluation data? If not, please indicate data not shown.
3.3, page 11, line 6, “Adjusting saliva concentration'', it is difficult to adjust the concentration of saliva in humans, so I feel that an expression such as Changing is more suitable (only English expression)
Author Response
<Introduction>
(1) Since it is a general, clearly indicate the purpose with the experiment
Response: Thanks for your good suggestion. The purpose of the experiment has been added.
<Method>
(2) How many repetitions of 2.3 & 2.4 ? Describe the number of repetitions that can explain the statistical significance.
Response: The number of repetitions was shown in 2.3. Six replicates were prepared for each comparison sample. Each set of experiments was measured at least six times.
(3) What kind of fibers are used in SPME ? References are not readily available for me, so it is a good to include them
Response: The details of the fibers have been added. The fibers were made by divinylbenzene, carboxen, and polydimethylsiloxane purchased from Supelco.
(4) Regarding the quantitative analysis method in 2.5, describe the method in more detail.
Response: The details of the quantitative analysis method have been added. The aroma compounds quantitation method was referred to in previous studies. Details can be also found in Reference 14.
Huang, X.-H.; Luo, Y.; Zhu, X.-H.; Ayed, C.; Fu, B.-S.; Dong, X.-P.; Fisk, I.; Qin, L. Dynamic release and perception of key odorants in grilled eel during chewing. Food Chemistry 2022, 378, 132073, doi:https://doi.org/10.1016/j.foodchem.2022.132073.
(5) In 2.6, clearly describe what kind of analysis you performed.
Response: The details of what kind of analysis was performed have been added.
<Results>
(6) Is the “the “number” of volatile components released'' in line 5 of 3.1 a mistype of “amount''? Figure 1 is data based on quantity, so if it is a number, it is better to indicate such as data not shown.
Response: We are very sorry for the misunderstanding caused by mistyping. The sentence has been revised.
(7) Figure 1, no statistical treatment was done, and it is unclear whether the difference is significant. Describe the number of times, statistical processing method, significance level in legend.
Response: The difference analysis was carried out in Figure 1. The difference level was represented by *. The significance level: * indicated p<0.05; ** indicated p<0.01; *** indicated p<0.001.
(8) 3.1 Line 5 from the end, "the enzymes in saliva mainly act on the tissue structure of fish meat [7]", is the citation appropriate?
Response: The citation was a review of saliva. Although the paper showed the view that enzymes in saliva can affect the structure of food during mastication, it was not directly related to the structure of the meat. To be more consistent with the view, a reference more relevant was added.
Qian, S.; Liu, K.; Wang, J.; Bai, F.; Gao, R.; Zeng, M.; Wu, J.; Zhao, Y.; Xu, X. Capturing the impact of oral processing behavior and bolus formation on the dynamic sensory perception and composition of steamed sturgeon meat. Food Chemistry: X 2023, 17, 100553.
(9) 3.1, page 5, line 2 from the end and 3.3, page 13, line 3 from the end, "the supramolecular complex" simply seems to be more appropriate as " bolus ".
Response: Thanks for your good suggestion. The word has been revised.
(10) 3.2, page 8, line 6 from the bottom, ”Consumption of grilled eel meat with a small amount of water may be able to effectively maintain the release of these compounds and help the aroma perception . ” How about quoting or describing the basis, and it will become clear whether 25% is a realistic set value
Response: Thanks for your good suggestion. Relevant references have been added. Saliva dilution information has been added to the paper.
(11) Figure 4. Conditions other than chewing time are unknown, so please describe them in legend
Response: Thanks for your suggestion. Details have been added to the legend.
(12) 3.3, page 10, last line, ”The increase in acid compounds not only changed the aroma perception of grilled eel during eating but also its taste.” You mention the taste, but do you have sensory evaluation data? If not, please indicate data not shown.
Response: We are sorry for the misunderstanding caused by the lack of clarity of expression. The sentence has been revised.
(13) 3.3, page 11, line 6, “Adjusting saliva concentration'', it is difficult to adjust the concentration of saliva in humans, so I feel that an expression such as Changing is more suitable (only English expression)
Response: Thanks for your suggestion. The sentence has been revised.

Reviewer 3 Report
The title implies that different foods are going to be included. It must include a reference to the fact that the study is carried out on grilled eel.
Figures 2 and 3 should have better resolution. It is not clear what each curve corresponds to. The font size of the axes is small.
In reference 21 the year does not appear in bold.
It would be interesting to include a brief mention in the discussion of how the different composition of saliva can modify food intake, in this case grilled eel.
Author Response
Response to Reviewer #2
1.The title implies that different foods are going to be included. It must include a reference to the fact that the study is carried out on grilled eel.
Response: Thanks for your good suggestion. The title has been revised.
“The Interaction Relationship of Aroma Components Releasing with Saliva and Chewing Degree during Grilled Eels Consumption”
2.Figures 2 and 3 should have better resolution. It is not clear what each curve corresponds to. The font size of the axes is small.
Response: Thanks for your good suggestion. The resolution size of the graph in the paper may be compressed due to the submission system. The resolution of Figures 2 and 3 has been improved. The font size of the axes has been increased.
3.In reference 21 the year does not appear in bold.
Response: Thanks for your good suggestion. The references has been checked and revised.
4.It would be interesting to include a brief mention in the discussion of how the different composition of saliva can modify food intake, in this case grilled eel.
Response: Thanks for your good suggestion. The discussion has been added.

Reviewer 4 Report
The authors studied the effect of saliva components and chewing time on odorants released with grilled eel meat. This research shows important information related to the chewing process of grilled eel meat. Overall, after thoroughly reading the paper, I recommend publishing the paper after a small revision.
1. Title: The title does not appropriately reflect the research conducted. Please, revise the title. The food under study should be included.
2. Introduction: The study was carried out on grilled eel meat, and the authors should highlight the importance and why this food was selected.
3. Materials and Methods: All these experiments were interesting and important information was obtained. However, it is necessary to include the number of samples analyzed, the statistical analysis performed, the means comparisons, etc.
4. Results and discussion: In the discussion section, please link the empirical results with a broader and deeper literature review.
The authors state: “The total amount of odors released by the control samples was higher than…”, “It was also higher than…”, “it was found that the release of volatiles decreased…”, “The amount released was greater than…”, etc. These types of statements must be based on statistical data analysis.
5. Fig 1: The figure is blurred and not easy to read. The quality of this figure needs to be improved.
6. Fig 2 and Fig 3:Please specify the number of samples analyzed, the meaning of the bars in the averages.
7. Fig 6. Please include the meaning of the colors and symbols *, **
The overall level of English in this manuscript is fine, only the details need to be revised.
Author Response
Response to Reviewer #3
1. Title: The title does not appropriately reflect the research conducted. Please, revise the title. The food under study should be included.
Response: Thanks for your good suggestion. The title has been revised.
“The Interaction Relationship of Aroma Components Releasing with Saliva and Chewing Degree during Grilled Eels Consumption”
2. Introduction: The study was carried out on grilled eel meat, and the authors should highlight the importance and why this food was selected.
Response: Thanks for your good suggestion. The information on grilled eels has been added. The importance of why selecting grilled eels has been explained.
“Traditional food, bearing the food culture and regional characteristics, has been concerned around the world. Each country has long been committed to the modernization of traditional food processing. As one of the typical traditional fish products, grilled eels attract consumers with their unique flavor. The sensory perception changes of consumers to the flavor quality of grilled eels have always been the focus of its process modernization.”
3.Materials and Methods: All these experiments were interesting and important information was obtained. However, it is necessary to include the number of samples analyzed, the statistical analysis performed, the means comparisons, etc.
Response: Thanks for your good suggestion. Detailed information on these experiments has been added. Six replicates were made for each sample. The bars in the figure are error lines. It was conducted by analyzing the difference of the measured values between different replicates. Microsoft Office 2016 (Microsoft, China), XLSTAT 2019 (Addinsoft, France), and SPSS v 24.0 (IBM, USA) were used for analyzing data, drawing figures, and combining figures. The statistical differences between different data were determined by SPSS v 24.0. Boxplot was made via XLSTAT 2019.
4.Results and discussion:In the discussion section, please link the empirical results with a broader and deeper literature review.
The authors state: “The total amount of odors released by the control samples was higher than…”, “It was also higher than…”, “it was found that the release of volatiles decreased…”, “The amount released was greater than…”, etc. These types of statements must be based on statistical data analysis.
Response: Thanks for your good suggestion. The discussion has been added.
5.Fig 1: The figure is blurred and not easy to read. The quality of this figure needs to be improved.
Response: The resolution size of the graph in the paper may be compressed due to the submission system. The resolution of Fig. 1 has been improved.
6.Fig 2 and Fig 3:Please specify the number of samples analyzed, the meaning of the bars in the averages.
Response: Thanks for your good suggestion. Six replicates were made for each sample. Details have been added in the Materials and Methods. The bars in the figure are error lines. It was conducted by analyzing the difference of the measured values between different replicates.
7.Fig 6. Please include the meaning of the colors and symbols *, **
Response: Thanks for your good suggestion. Each colored cell on the heat map corresponds to correlation value between aroma compounds and chewing, enzyme, or inorganic ions. The significance level: * indicated p<0.05; ** indicated p<0.01. Details have been added in the Figure 6 legend.
8.Comments on the Quality of English Language
The overall level of English in this manuscript is fine, only the details need to be revised.
Response: The grammar and English expression of the paper have been checked and revised.

Round 2
Reviewer 1 Report
After a thorough review of the author's work, I am happy to report that all of my observations, comments, and suggestions have been addressed. The authors have done an outstanding job of responding to my concerns, and I fully support their responses. It is my belief that they have taken the necessary measures to guarantee the accuracy and dependability of their research.
Author Response
Thank you for your suggestion for the manuscript. We are glad to your satisfaction with the revised manuscript. Meanwhile, we have checked the manuscript again and tried to improve it to the best.